# The Local Microbiome in Esophageal Cancer and Treatment Response: A Review of Emerging Data and Future Directions

**DOI:** 10.3390/cancers15143562

**Published:** 2023-07-10

**Authors:** Abhishek Pandey, Christopher H. Lieu, Sunnie S. Kim

**Affiliations:** 1University of Colorado School of Medicine, Aurora, CO 80045, USA; 2Department of Medicine, Division of Medical Oncology, University of Colorado Anschutz Medical Campus, Aurora, CO 80045, USA

**Keywords:** microbiome, esophageal cancer, gastric cancer, immunotherapy

## Abstract

**Simple Summary:**

Esophageal and esophagogastric junction cancers are not well explained by traditional risk factors and have high mortality rates and limited treatment options. The aim of this review article is to describe the potential role of the local tumor microbiome in upper GI carcinogenesis, as well as treatment responses, and to further describe the challenges and opportunities in conducting and interpreting high-quality microbiome studies. Immunotherapy is now approved for clinical use, and several lessons from studies of the gut microbiome’s role in immune checkpoint inhibitor responses in melanoma, lung, and genitourinary cancers may also inform esophageal cancer studies.

**Abstract:**

The incidence of esophageal cancer is increasing worldwide, with established risk factors explaining only a small fraction of cases. Currently, there are no established screening protocols in most countries, and treatment options are limited. The human microbiome has been implicated in carcinogenesis and the cancer treatment response. The advent of nucleic acid sequencing technologies has enabled more comprehensive, culture-independent bacterial identification. Across several tumor types, studies of tissue-specific microbiomes have shown associations between the overall microbiome composition, the relative abundance of specific bacteria, and tumorigenesis. Furthermore, in the era of cancer immunotherapy, several studies have demonstrated that the microbiome and specific bacteria may modify treatment responses and the risk of immune-related adverse events. Design: peer-reviewed, published studies describing the role of local, gastrointestinal-specific microbiota or the role of the gut microbiome in treatment responses were reviewed. PubMed was searched from 1 September 2022 to 1 November 2022, using the following terms in combination: “microbiome”, “tumor microbiome”, “esophageal cancer”, “cancer”, “cancer treatment”, and “immunotherapy”. Original research articles were considered, and other reviews or editorials were discarded. In total, approximately 250 articles were considered. Results: over 70 studies describing microbiome research in either gastrointestinal carcinogenesis or the systemic treatment response were identified and reviewed. Conclusions: a growing body of evidence supports the role of the esophageal microbiome in both esophageal tumorigenesis and the immune checkpoint inhibitor response. More well-designed, comprehensive studies are required to collect the appropriate clinical, microbial, and immunophenotype data that are needed to clarify the precise role of the microbiome in esophageal carcinogenesis and treatment.

## 1. Introduction

The incidence of esophageal cancer, specifically adenocarcinoma, in the US has been rising over the past several years. The majority of esophageal cancers are not explained by germline DNA mutations, and, while traditional risk factors such as tobacco use and alcohol consumption have declined, trends in cancer incidence are mixed. Smoking cessation is associated with decreases in esophageal squamous cell cancers (ESCC), but the incidence of esophageal adenocarcinoma (EAC) has increased slightly over a similar timeline.

A growing body of evidence shows that the development of EAC is a multi-step, multi-factorial process, with much still unknown. Obesity is linked to EAC, and a meta-analysis showed that a BMI greater than 30 kg/m^2^ is associated with higher risk of developing EAC. While gastro-esophageal reflux disease (GERD) is often associated with obesity, only 22% of obese patients developed GERD, compared to 14% of non-obese patients^1^. GERD can lead to Barrett esophagus (BE), a known pre-cancerous condition, but only 0.3% of BE cases progress to EAC annually [1]. 

EAC treatment now includes cytotoxic chemotherapy, radiotherapy, HER2-directed antibodies, and, more recently, immune checkpoint inhibitors (ICI), which are now approved as part of the treatment regimen for locally advanced and metastatic ESCC and EAC [2,3]. While ICIs have improved survival outcomes for patients, there is significant variability in the treatment response across patients, with no known reliable predictive biomarker [4]. The tumor mutational burden and immune cell signatures such as programmed death ligand 1 (PD-L1) are imperfect predictors of the ICI response. More recently, the human microbiome (MB) has been hypothesized as a potential modifier of the ICI response. 

We aim to review the role of the MB in esophageal carcinogenesis and treatment. We briefly describe MB research methods and then review the relevant studies, across esophageal, other gastrointestinal (GI), and non-GI tumors, which support the role of the MB in modifying responses to cancer treatment, with a focus on immunotherapy (Figure 1). Finally, we review the challenges of conducting MB studies of the upper GI tract and describe future research directions.

## 2. Defining “Microbiome”

The term “microbiome” (MB) describes a community of bacteria, viruses, fungi, and other microbes inhabiting a tissue or organ system, with bacteria making up the vast majority of the biomass. Bacteria within the “healthy” human GI tract are commensal, contributing to the digestion of complex carbohydrates, vitamin and amino acid synthesis, and drug metabolism [5]. Commensals also provide a physical “barrier” against pathogens, bacteria known to cause disease when present, and pathobionts, or bacteria that behave as pathogens only if their numbers increase above a certain threshold [6,7]. The GI MB is typically in a healthy and steady state, but, if altered to a prolonged state of dysbiosis, it may contribute to the pathology of several cardiovascular [8], metabolic [9], neurologic [10], and malignant diseases. 

## 3. Detecting the Microbiome

Early culture-based studies estimated that the entire GI tract had 400–500 species, but more recent DNA-based analyses now estimate that this number is closer to 15,000 species [11]. Polymerase chain reaction (PCR)-amplified 16S ribosomal RNA (rRNA) sequencing is a thoroughly validated method for genus- and species-level identification. Because all bacteria have an rRNA gene, this region can be targeted reliably. 

“Next-generation sequencing” (NGS) has enabled “shotgun” metagenomic sequencing (metagenomics), interrogating all sequenced reads from a sample, and filtering out non-MB (i.e., human) DNA. Metagenomics goes beyond the 16S rRNA region, and the identified microbial genes can be mapped to microbial reference genomes. Genomic analyses can be combined with the MB-wide RNA expression of bacterial genes (metatranscriptomics) and mass spectrometry methods to identify MB-associated bile acids, short-chain fatty acids, amino acid metabolites, and polysaccharides (metabolomics). This multi-omics approach is often used to describe MB–MB or MB–host signaling interactions. 

The sites for human MB sampling can pose different challenges. The esophagus is a relatively sterile “low-biomass” environment compared to, for example, the colon [12]. While high-biomass fecal samples may be easier to obtain and analyze, they more closely reflect the distal GI tract and should not be used in studies of the upper GI. Oropharyngeal MB samples have demonstrated the MB’s overlap with esophageal samples [13,14], but there is significant variability between the oral mucosal and esophageal mucosal MB [15,16], and they should not be considered substitutes for direct esophageal sampling via a core needle biopsy or resection. Later in the review, we describe the emerging esophageal sampling methods that may constitute alternatives or supplements to tissue biopsies. 

## 4. Describing the Microbiome

In MB research, alpha diversity (a-diversity) measures intra-sample diversity and represents the distribution and abundance of unique bacterial taxa (i.e., order, genus, species, strain) within a sample. For example, a human fecal sample with high a-diversity is composed of a more diverse community of bacteria compared to a sample with lower a-diversity. Common conditions associated with low a-diversity states are inflammatory bowel disease [17], gastrointestinal infections, and antibiotic use [18]. 

Inter-sample diversity is measured by beta-diversity (B-diversity). It describes the stability or volatility in bacterial composition over time or across samples within a cohort. For example, an individual with high B-diversity would have an MB that is more volatile over time and/or serial sampling compared to an individual with low B-diversity between repeated samples. Patients with IBD have demonstrated higher B-diversity compared to those without IBD [18].

MB studies may report bacterial identity in terms of “clusters”, grouping phylogenetically similar taxa (i.e., phyla, species) together to stratify different groups by a functional outcome (i.e., an association with the immune response, tumor growth, or survival). Moreover, 16S rRNA results are often reported as “operational taxonomic units” (OTUs), which typically represent sequences that are not identical but are similar enough to be considered the same genus or species [19]. Metagenomics analyses often describe the presence of other genes within a functional “network analysis” that includes gene expression and metabolite data [20].

## 5. Esophageal Microbiome

In 2004, Blaser et al. were one of the first groups to report a 16S rRNA PCR analysis of histologically normal tissue biopsies from the distal esophagi of healthy patients. Along with several other groups, they characterized the healthy esophageal MB as having a high relative abundance of *Streptococcus*. They found that a transition to esophagitis and/or Barrett’s esophagus correlated with a dysbiosis dominated by *Bacteroides*, *Proteobacteria*, *Fusobacteria*, *Firmicutes*, and *Campylobacter* [21,22,23,24,25].

In 2016, Yamamura et al. analyzed FFPE from 300 primary ESCC and 12 EAC, along with paired adjacent normal tissue, from surgically resected specimens. Using qPCR to amplify the nusG gene of *F. nucleatum*, they detected *F. nucleatum* in 23% (74) of cases. Samples enriched for *F. nucleatum*, compared to those without qPCR-detected bacteria, correlated with statistically significantly shorter survival. Notably, the presence of *F. nucleatum* was not associated with the subject’s age, sex, smoking status, alcohol consumption status, tumor size, tumor histology, the location of the tumor, or pre-operative/neo-adjuvant chemotherapy or radiotherapy. The effect of *F. nucleatum* was not significantly modified by age, performance status, tumor location, preoperative treatment, or the tumor size or stage [26]. 

Deshpande et al. used a multi-omics approach to show that the local MB plays a role in the progression of GERD to esophageal metaplasia. They prospectively recruited 47 patients who met the endoscopic criteria. In total, 13 had GERD, 7 had metaplasia, and the remaining 27 were histologically normal, serving as controls. There were no significant differences in BMI or prior PPI use, and the use of antibiotics or NSAIDs within two months was an exclusion criterion. These researchers further supplemented their study with a previously published cohort of 100 cases (GERD, n = 29; metaplasia, n = 12; normal, n = 100). 

Compared to normal and GERD samples, the metaplastic specimens had an abundance of *Campylobacter* species, a known relative of the gastric carcinogen *Helicobacter pylori*. MB RNA-Seq showed that metaplastic transcriptomes were more similar to intestinal columnar cells and colorectal adenocarcinoma than normal or GERD MB transcriptomes. Pathway analysis showed the upregulation of bile acid secretion, cAMP signaling, lysosomal processing, and several other networks associated with carcinogenesis. Healthy, GERD, and metaplastic esophageal tissue transcriptomes corresponded to changes in the MB composition, though the significance of the correlation is unclear. It may be that the association seen was driven mostly by the relative abundance of *Campylobacter*. 

*Campylobacter* is an intracellular microbe, and the identified *Campylobacter* strains were isolated and co-cultured with allogeneic human macrophages. Cytokine release assays of infected macrophages demonstrated persistent inflammation (up to 18 h) compared to non-infected macrophages. Interestingly, different *Campylobacter* strains had varying inflammatory effects and varying intracellular survival potential. *C. rectus*, in particular, had a longer in vitro survival time, and, when cultured with normal esophageal epithelial cells, demonstrated the host upregulation of several genes that were overexpressed in the analyzed metaplastic samples, including PIGN, a chromosomal instability suppressor [16].

A potential mechanism of esophageal carcinogenesis involves toll-like receptor (TLR) signaling, which allows epithelial cells to recognize luminal pathogens and their components, such as lipopolysaccharides, flagellins, DNA, and RNA. This triggers an innate immune response and inflammation. Alterations in cytokine and chemokine expression can lead to angiogenesis and potential tumorogenesis. High expression levels of TLR-5 has been associated with the progression from metaplasia/dysplasia to esophageal adenocarcinoma [27].

## 6. The Microbiome in the Carcinogenesis of Other Gastrointestinal Cancers

While several studies of the esophageal MB provide correlative data, there are significantly fewer data available at present that would allow us to establish causality. Non-esophageal studies have made more progress and implicated dysbiosis, as well as several bacteria, as potential carcinogens. Whether MB changes are more likely to be a “driver” of dysplasia and neoplasia or a “passenger”, secondary to carcinogenesis, remains to be demonstrated. The current data support a correlative association with certain bacteria, demonstrating potential causal mechanisms of gastrointestinal cancers (Figure 2). 

*H. pylori* is arguably the most well-known bacterial carcinogen, contributing to gastric cancers and mucosa-associated tissue lymphomas (MALT) [28,29,30]. It attaches directly to gastric epithelial cells via HopQ binding to host CEACAM, allowing the bacteria to inject CagA and activating Wnt/beta-catenin signaling, a known neoplastic process [31]. As described earlier, *Campylobacter* is phylogenetically similar to *Helicobacter* and may share similar inter-cellular signaling mechanisms. Gastric mucosal biopsies from intra-epithelial neoplasia and adenocarcinoma patients have shown a higher abundance of *Actinobacteria*, *Bacteroides*, *Firmicutes*, and *Fusobacteria* [32]. 

Nakatsu et al. studied paired adenoma and adenoma-adjacent normal colonic mucosa, as well as adenocarcinoma and adjacent normal tissues, to show that histologically normal lower-GI mucosa is heavily populated with *Lactobacilli*, *Bacteroides*, and *Bifidobacterium* species, whereas adenocarcinoma has a distinct dysbiosis with a relative abundance of *Fusobacterium*, *Gemella*, *Porphyromonas*, and *Parvimonas*—the latter two being well-known oral commensals. Overall, adenocarcinoma samples showed less a-diversity than adenoma or normal tissue, suggesting that a small subset of bacteria may be more pathogenic than the overall community [33]. Fecal sampling studies have further implicated certain *Enterococcus*, *Shigella*, and *Klebsiella* species in colorectal carcinogenesis [34]. 

*Fusobacterium nucleatum* has been implicated as a carcinogen in oral squamous cell carcinoma (OSCC) and CRC, with a demonstrated bacterial transcript load 400 times that of matched adjacent normal tissue [35]. *F. nucleatum* enters host cells and is associated with NF-KB and IL-6 upregulation in OSCC cells [36], and with IL-8- and TNF-alpha-mediated inflammation in both cancer types [37]. 

Mouse models have demonstrated that the pancreas is relatively sterile, and that bacterial translocation from the intestine can lead to carcinogenesis. Human pancreatic ductal adenocarcinoma samples have been shown to be enriched for *Proteobacteria*, *Bacteroides*, and *Firmicutes* compared to normal pancreatic tissue [38]. Several studies have shown that the liver is susceptible to both bacterial translocation from the intestine and the portal venous transport of active gut bacterial metabolites [39,40]. 

*Escherichia coli* generates and secretes colibactin, which crosslinks eukaryotic DNA, leading to double-strand DNA breaks [41,42]. *Salmonella enterica* activates the MAPK-AKT pathway in mouse and cell-line models of gallbladder cancer [42]. Certain strains of *Bacteroides fragilis*, a known intestinal symbiont, have been shown to up-regulate Wnt/B-catenin and NF–KB signaling in chronic colitis and CRC tissue. These strains secrete a specific exotoxin, associated with increased pro-inflammatory Th17 T-cell activity, promoting cell survival [43]. 

As described above, any MB-mediated carcinogenesis likely occurs via direct cytotoxicity and chronic inflammation, DNA damage, and the dysregulation of the cell cycle and apoptotic pathways. While outside the scope of this review, it is worth noting that the human “virome” (i.e., human papilloma virus, other eukaryotic viruses, and bacteriophages) [44,45], the human “mycobiome” (i.e., fungus) [46], and the parasitome [47] may have different and unique mechanisms of carcinogenesis. 

## 7. Mb Signaling, Immune Cell Interactions & Treatment Responses

On a functional level, several MB-derived metabolites interact with tumor cells, immune cells, and even therapeutic agents. Kalaora et al. showed that melanoma cells can present MHC-restricted bacterial fragments, eliciting T-cell activity against the antigen-presenting cell [48]. Certain commensal bacteria synthesize the metabolite inosine, which may enhance the efficacy of ICI therapy [49]. Bacterially derived short-chain fatty acids (SCFA) such as acetate, propionate, and butyrate can modify immune cell activity [50,51]. These SCFAs can promote regulatory T-cell (Treg) expansion, as well as increasing effector T-cell activity [52]. Butyrate is associated with protection against graft-versus-host disease [53] and may promote the memory potential and antiviral cytotoxic effector functions of CD8+ T-cells [54].

### 7.1. Chemotherapy

Animal models have demonstrated a dynamic relationship between chemotherapy and the gut MB. Germ-free mice and antibiotic-treated mice have both demonstrated decreased oxaliplatin toxicity and increased tumor growth [55]. Gellar et al. showed that the bacterial enzyme cytidine deaminase (CD) metabolized gemcitabine into an inactive form. They used a CD-producing *Gammaproteobacteria* species to induce gemcitabine resistance in a colon cancer model [56]. Antibiotic pre-treatment before cyclophosphamide infusion blunts the immune cell response, allowing for tumor growth [57]. Conversely, chemotherapy may be synergistic with gut commensal bacteria. Chemotherapy-induced gut epithelial damage allows bacterial translocation to local lymph nodes, where Th17 and cytotoxic T-cells are activated [58]. Other studies have shown differences in pre- and post-chemotherapy gut MB a-diversity [59].

### 7.2. Immunotherapy

The MB interacts with the innate and adaptive immune responses in various ways. Microbe-associated molecular patterns (MAMPs) on commensal or pathogenic bacteria cell surfaces are sensed by neutrophils, macrophages, and NK cells, which may explain their activity as cancer vaccine adjuvants [60]. The gut MB regulates the tryptophan metabolite 5-HT [61], which binds T-cell aryl hydrocarbon receptors, leading to the upregulation of several inhibitory markers and, eventually, T-cell exhaustion [62], a phenomenon associated with ICI treatment failure. The first studies to demonstrate the gut MB modification of ICI therapy responses were mouse models showing *Bifidobacterium* species to be associated with increased anti-PD-L1 activity [63], and *Bacteroides thetalotaomicron* and fragilis with increased anti-CTLA-4 efficacy [64]. Both studies showed increased T-cell maturation and cytokine activity, along with decreased tumor growth.

With the widespread adoption of ICIs in melanoma treatment, several groups have explored the role of the gut MB in modifying treatment responses (Table 1).

Chaput et al. [65] performed baseline and serial fecal 16S rRNA sequencing, along with peripheral blood T-cell immunophenotyping and cytokine profiling, prior to 4 ipilimumab infusions in 26 patients with metastatic melanoma. They found that the baseline gut MB composition differed significantly between responders and non-responders, although their alpha-diversity was equivalent. Non-responders had a higher relative abundance of *Bacteroides* species. Responders were enriched for the Firmicutes and Faecalibacterium populations. Responders with predominantly *Faecalibacterium*-enriched stools had a lower baseline proportion of peripherally circulating Tregs and alpha-4/beta-7-CD4þ T-cells, a subtype with known effector and memory functions. Baseline peripheral blood cytokine signatures did not correlate with treatment response or any particular bacterial community. Interestingly, the gut MB composition did not significantly change over the course of treatment in responders and non-responders alike, but there were significant reductions over time in the a-diversity of samples from seven patients who experienced immune-related colitis.

Frankel et al. [66] analyzed fecal samples from 39 patients with metastatic melanoma prior to Ipilimumab monotherapy, nivolumab monotherapy, a combination of Ipilimumab and nivolumab, or pembrolizumab. They used metagenomic shotgun sequencing methods supplemented with fecal metabolomic analysis. They did not analyze peripheral blood immunophenotypes. Among all ICI-treated patients, there was no significant difference in alpha-diversity between responders and responders. Responders had a higher relative abundance of *Bacteroides caccae* and *Streptococcus parasanguinis* compared to non-responders. A functional analysis of MB genes showed that ICI responder MB genomes were enriched for fatty acid synthesis. Interestingly, pro-inflammatory anacardic acid levels were increased in ICI-responder fecal samples. Anacardic acid is not a known bacterial product or metabolite but is found in cashews. While the authors were collecting the antibiotic and probiotic histories, they found that that the six highest anacardic acid levels were associated with near-daily cashew consumption. Antibiotic or probiotic use was limited to only five patients, with no significant MB associations.

Matson et al [68]. examined fecal samples from 42 metastatic melanoma patients prior to anti-PD-L1 and/or anti-CTLA4 therapy using a sequenced combination of 16S rRNA, metagenomics, and quantitative PCR (qPCR) for known species with validated primers. This stepwise method had the advantage of screening for bacteria that can be reliably identified at the species level, with the disadvantage of being dependent on validated primers for known bacteria. They found eight species to be relatively more abundant in responders: *Enterococcus faecium*, *Collinsella aerofaciens*, *Bifidobacterium adolescentis*, *Klebsiella pneumoniae*, *Veillonella parvula*, *Parabacteroides merdae*, *Lactobacillus* sp., *and Bifidobacterium longus*. Two species were more abundant in non-responders: *Ruminococcus abeum* and *Roseburia intestinalis*. FMTs from responding patients to germ-free mice showed improved tumor control with anti-PDL1 therapy.

Gopalakrishnan et al [67]. analyzed fecal samples from 112 metastatic melanoma patients prior to ICI therapy and 6 months after the initial ICI infusion. They also sampled the oral MB and peripheral blood immunophenotypes, as well as tumor biopsies, for genomic analysis, in order to control for a higher tumor mutational burden, a favorable biomarker for ICI response. They found the oral and fecal/gut MB to be distinct communities, and responders and non-responders had “comparable” mutational burdens and melanoma-specific driver mutations.

They found that the responders’ fecal MB had higher a-diversity than that of non-responders, with no significant differences seen in oral MB a-diversity between the groups. Notably, they found that the highest third of the samples in terms of a-diversity correlated with better ICI treatment responses compared to the lower two-thirds of samples. Metagenomics showed that responders were enriched for *Ruminococcaceae* and two *Faecalibacterium* species. A cohort of 19 patients with a high abundance of a *Faecalibacterium* species did not reach median progression-free survival, while a cohort of 20 patients with low abundance had a median PFS of 242 days. The authors did not describe their methods for separating “high” and “low” abundance. B-diversity was reported, with responders and non-responders having distinctly different patterns of changes in community composition before and after treatment. A metabolic pathway network analysis of the metagenomic data showed differences in “biosynthesis” and “degradation” genes between responders and non-responders. Responders were more likely to have bacterial genes associated with the synthesis and breakdown of fatty acids and amino acids and with the breakdown of bile acids.

An intriguing part of the study conducted by Gopalakrishnan et al. [67] concerns their analysis of tumor-associated tumor-infiltrating immune cells (TILs) and MB. In general, responders had a higher density of CD8+ T-cells. A higher relative abundance of *Faecalibacterium* and *Ruminococcaceae* had a statistically significant correlation with CD8+ T-cell density within the tumor. These patients also had larger populations of peripherally circulating effector CD4+and CD8+ T-cells, with a strong in vitro anti-PD1 response. Conversely, a higher relative abundance of *Bacteroides* species was associated with a larger population of peripherally circulating immunosuppressive Tregs and “myeloid-derived suppressor cells”. *Bacteroides*-enriched patients also showed a negative correlation with tumor-associated CD8+ TILs. Finally, the transplantation of human responders’ fecal matter into germ-free mice showed improved ICI responses and tumor control.

Building on this prior work, Tanuoe et al. [70] demonstrated that 11 specific bacterial strains, consisting of *Ruminococcacae*, *Paraprevotella*, *Fusobacteria*, *Alistipes*, and *Bacteroides* species from healthy human donor feces, could induce activity in colon-specific IFNy+ CD8+ T-cells in vitro and in the colons of germ-free mice. Furthermore, over time, the bacterial mix could recruit more T-cells to the colon, induce proliferation, and maintain elevated T-cell activity more than one week after inoculation. Interestingly, increased numbers of IFNy+ CD8 effector T-cells were also found in mesenteric lymph nodes, the liver, and lung tissues, supporting a systemic effect as well as a local colonic immune response. However, the extra-colonic T-cells had different characteristics and levels of activity, suggestive of T-cell differentiation or evolution in the periphery. Finally, in a tumor-engrafted mouse model, they showed that the 11-strain mixture could improve the control of tumor growth in the absence of other treatments, and, when inoculation occurred prior to anti-PD-1 treatment, enhanced the ICI effect.

ICI responses outside of melanoma have been studied as well. Routy et al. [69] examined 249 patients with metastatic non-small-cell lung cancer (NSCLC), renal cell carcinoma (RCC), or urothelial carcinoma, 69 of whom used antibiotics within the 2 months prior to their first ICI infusion or 1 month afterwards. They found that antibiotic-treated patients had a significantly shorter progression-free survival (PFS) period and overall survival (OS) interval compared to the untreated subgroup, and this association was independent of age, performance status, or other prognostic measures. A follow-up validation cohort of 239 patients with metastatic NSCLC confirmed the initial negative antibiotic impact on PFS and OS. Metagenomics showed that ICI responders had a higher relative abundance of *Firmicutes*, *Akkermansia*, and *Alistipes* species compared to non-responders. Peripheral blood T-cell activity assessed by IFN-release assays did not correlate with the fecal MB composition. Finally, to study the causal relationship, the fecal matter from ICI-responding and non-responding humans, respectively, were transplanted (FMT) into germ-free or antibiotic-pretreated mice prior to the anti-PD1 treatment. Responder FMTs were associated with a better anti-tumor response than non-responder FMTs.

### 7.3. Cellular Therapy

Luu et al. [73] demonstrated that treatment with the short-chain fatty acids pentanoate and butyrate induces the strong production of effector molecules in cytotoxic CD8+ T-cells and CAR-T cells, resulting in increased anti-tumor reactivity and improved therapeutic outcomes. Similarly, Smith et al. also saw an association with the butyrate-producing *Faecalibacterium* in the gut microbiome and increased responses to CD19 CAR T-cell therapy in high-risk hematologic malignancies [74]. Lower butyrate concentrations have been shown to increase regulatory T-cells, while higher butyrate concentrations mediate IFN-γ-producing regulatory T-cells or conventional T-cells, improving the anti-tumor response [51,52]. These data show that specific gut-bacteria-derived metabolites, such as butyrate and pentanoate, have the potential to optimize adoptive T-cell therapy for cancer in humans.

### 7.4. Fecal Microbiome Transplant

Baruch et al. [71] and Davar et al. [72] conducted the first-in-human trials showing that FMTs from ICI-responsive metastatic melanoma patients salvaged ICI-nonresponsive metastatic melanoma patients. Baruch et al. saw responses in 3 of 10 patients, with the histological analysis of the intestinal lamina propria of FMT recipients showing increased TIL content. Davar et al. recovered the ICI response in 6 of 15 patients and showed that the donor FMT engrafted in a durable manner.

FMTs that resulted in clinical response were enriched with *Ruminococcaceae*,* Bifidobacteriaceae*,* Lachnospiraceae*, and *Erysiplotrichaceae*, while FMTs that were mostly populated with *Tannerellaceae*,* Sutterellaceae*, and *Bacteroidaeceae* were ineffective. Perhaps the most likely reason for the response was the higher proportion of peripheral CD8+ and activated mucosal-associated invariant T-cells, which both produce molecules such as granzyme, perforin, and NK-activating receptors.

## 8. Conclusions

Pre-clinical and clinical data support the human MB’s role in the carcinogenesis and modulation of anti-tumor host immune responses, as well as systemic chemotherapy, immunotherapy, and adoptive cellular therapy. Bacteria may act as carcinogens through direct cytotoxicity, DNA damage, or cell cycle dysregulation, or as mediators of proliferation or apoptosis. Bacterial metabolites may act both locally and peripherally to influence immune cell differentiation and activity. The MB may modify treatment responses via the chemotherapy metabolism and the modulation of local tumor-infiltrating and circulating peripheral innate and adaptive immune cells, as well as intra-tumoral MHC-restricted antigen presentation.

Most work to date has focused on melanoma, NSCLC, or colon cancers, with relatively sparse data describing the mechanisms of esophageal carcinogenesis or the MB modulation of therapy. Perhaps due to the relatively lower prevalence of esophageal cancer or the relatively recent use of immunotherapy in this disease setting, very little is known about the potential for prognostic, predictive, or modifiable biomarkers within this population.

A significant obstacle to esophageal MB research may be sample acquisition, processing, and analysis. Fecal matter is rich in microbes, and sampling is relatively straight-forward compared to esophageal biopsies, which, as described earlier, are relatively sterile and may contain less tissue and biomass for analysis. However, several alternatives to direct endoscopy and biopsy have recently been developed and validated for esophageal MB research. Esophageal “string testing” is a minimally invasive, capsule-based technology that has been used to sample the esophageal MB in pediatric populations [75]. Cytosponge, another non-endoscopic capsule-based technology validated for BE detection in the UK, has been used to sample the esophageal MB in BE patients [25]. Peripheral blood metagenomic profiling may constitute another non-invasive method for MB characterization in EAC patients [76]. These methods may be good supplements, if not substitutes, for tissue biopsies in future studies.

The role of the local esophageal MB in modifying cancer treatment responses remains unknown. The backbone of esophageal cancer treatment today is multi-modal, with chemotherapy, radiotherapy, and immunotherapy all playing a part. The local MB may play a role in modifying the response to any of these agents. The upper and lower GI tracts have different MB compositions, but they share a similar anatomy and lymphatic drainage patterns, suggesting that the MB–immune cell relationships seen in colon studies may apply to the esophagus and upper GI tract. Building on the studies described in this review, several groups are investigating FMTs, oral microbial treatments and specific probiotics in patients receiving ICIs, allogeneic stem cell transplants, chemotherapy, and radiotherapy (Table 2). However, none of these studies focus on or even include esophageal cancers in their study population.

While the gut microbiome and fecal sampling may not aid in describing local esophageal processes, several studies mentioned in this review (Table 1) implicate intestinal microbiota as modifying the ICI treatment response. With the recent addition of ICIs to the treatment of both locally advanced and metastatic upper-GI cancers, these study results may also be applicable in the upper-GI setting. The same MB–immune cell interactions present in the guts of ICI-treated melanoma or NSCLC patients may also be present in patients with upper-GI cancer.

Further investigations and more data are needed to understand the role of the local esophageal MB in carcinogenesis and treatment. Specifically, investigators should aim to interrogate pre-cancerous and cancerous cohorts to better describe longitudinal changes, potential modifiable biomarkers, and opportunities for cancer prevention. Clinical data describing antibiotic and other MB-altering drug use, as well as dietary histories, should be collected. The esophageal tumor microenvironment, tumor-infiltrating immune cells, the local esophageal MB, and local signaling interactions between the TME, MB, and immune cells, as well as the longitudinal evolution of the peripheral circulating immune cell response to immunotherapy, should all be investigated in future studies of esophageal cancers. We hope that future reviews on this topic will have more esophageal studies to discuss.

## Figures and Tables

**Figure 1 cancers-15-03562-f001:**
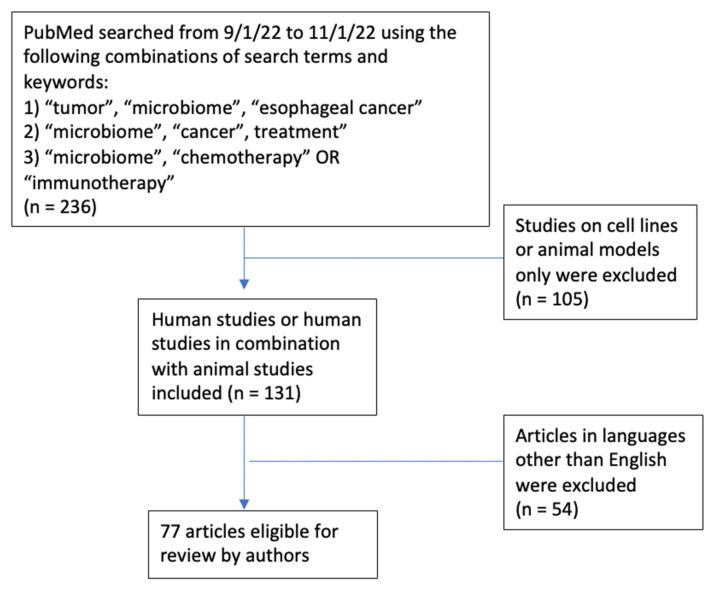
Schematic diagram.

**Figure 2 cancers-15-03562-f002:**
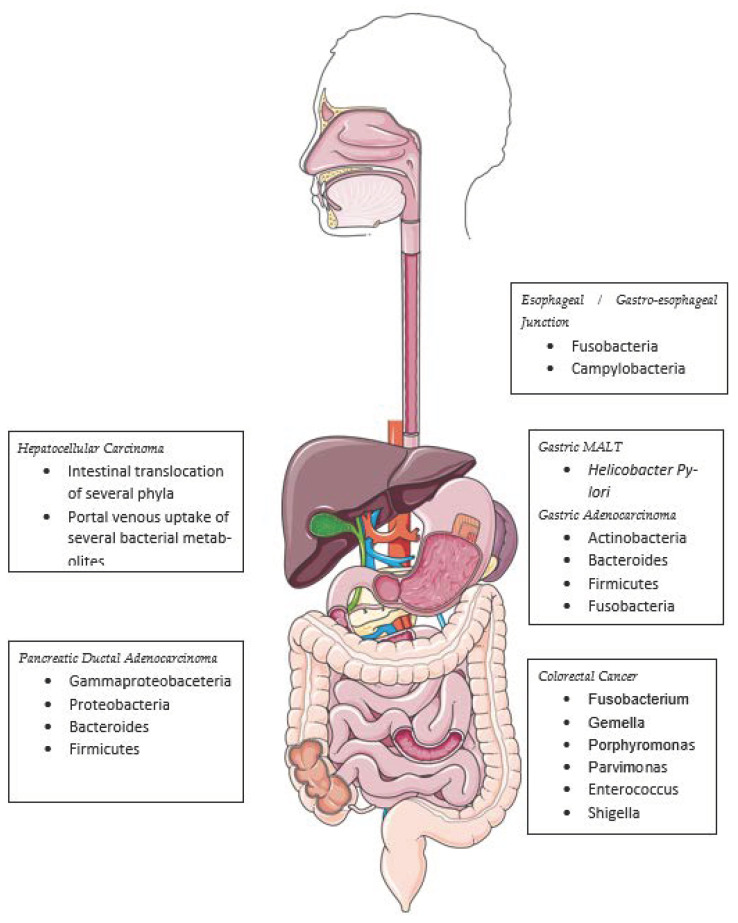
Bacteria implicated in gastrointestinal carcinogenesis.

**Table 1 cancers-15-03562-t001:** Overview of studies investigating role of the microbiome (MB) in the immune checkpoint inhibitor (ICI) response. CTLA-4 = cytotoxic T-lymphocyte-associated protein 4, PD-1 = Programmed cell death protein 1, PD-L1 = programmed death ligand 1, TILs = tumor-infiltrating lymphocytes, FMT = fecal microbial transplant.

Study	Tumor	Subjects	Immunotherapy	Methods	Findings
Chaput (2017) [65]	Metastatic Melanoma	Human	Anti-CTLA-4	Fecal 16S rRNA, Peripheral blood immunophenotyping	Responders enriched for *Faecalibacterium* and *Firmicutes*, and with lower baseline peripheral Tregs and alpha-beta CD4 T-cells
Frankel (2017) [66]	Metastatic Melanoma	Human	Anti-CTLA-4, Anti-PD-1, Anti-CTLA-4 + Anti-PD-1	Fecal metagenomics, fecal metabolomics	Responders enriched for *Bacteroides* and *Streptococcus*; MB enriched for fatty acid synthesis
Gopalakrishnan (2018) [67]	Metastatic Melanoma	Human, mouse	Anti-PD-1	Oral and fecal metagenomics; tumor mutation burden; tumor-infiltrating lymphocytes, peripheral blood immunophenotypes	Responders enriched for *Ruminococcaceae* and *Faecalibacterium*, and with greater CD8+ TIL density; *Bacteroides* abundance associated with non-response and fewer CD8+ TILs
Matson (2018) [68]	Metastatic Melanoma	Human, mouse	Anti-PD-1, Anti-CTLA-4	Fecal 16S rRNA, fecal metagenomics, fecal qPCR; FMT	Responders enriched for *E. faecium*, *C. aerofaciens*, *B. adolescentis*, *B. lonus*, *K. pneumoniae*, *V. parvula*, *P. merdae*; FMT from responders to germ-free mice with improved tumor control with anti-PD-1 therapy
Routy (2018) [69]	Non-small-cell lung carcinoma, Renal cell carcinoma/urothelial carcinoma	Human, mouse	Anti-PD-1/PD-L1	Fecal metagenomics; FMT	ICI responders enriched for *Firmicutes*, *Akkermansia*, *Alistipes*; germ-free mice with better anti-tumor response after human responder FMT than non-responder FMT
Tanoue (2019) [70]	MC38 adenocarcinoma mouse model	Human, mouse	Anti-PD-1	FMT of pre-specified mix of 11 bacterial strains	FMT-treated mice with better spontaneous (without ICI) and anti-PD-1 tumor responses
Baruch (2021) [71]	Metastatic Melanoma	Human	Anti-PD-1	Human-to-human, ICI-responsive-to-non-responsive FMT	3/10 non-responders salvaged anti-PD-1 tumor response
Davar (2021) [72]	Metastatic Melanoma	Human	Anti-PD-1	Human-to-human, ICI-responsive-to-non-responsive FMT	6/15 non-responders salvaged anti-PD-1 tumor response

**Table 2 cancers-15-03562-t002:** Overview of studies investigating the role of microbiome-based interventions. FMT: fecal microbial transplant, ICI: immune checkpoint inhibitor.

MB Intervention	Cancer Treatment	Tumor Type	Study Phase	Trial ID	Status
FMT	ICI	Renal cell Carcinoma/bladder	1	NCT04038619	Active, recruiting
FMT	ICI	Melanoma, non-small-cell lung cancer	1	NCT03819296	Active, recruiting
FMT	ICI	Melanoma, non-small-cell lung cancer	2	NCT04951583	Active, recruiting
FMT	ICI	Prostate	2	NCT04116775	Active, recruiting
FMT	ICI	Renal cell carcinoma	2	NCT04758507	Active, recruiting
FMT	Allogeneic stem cell transplant	Hematologic malignancies	2	NCT04935684	Active, recruiting
Probiotic	ICI + chemotherapy	Non-small-cell lung cancer	2	NCT04699721	Active
Probiotic	Chemotherapy	Colorectal cancer	2	NCT04131803	Active
Prebiotic/Fiber	ICI	Melanoma	2	NCT04645680	Active, recruiting
Prebiotic/Fiber	Radiotherapy	Colorectal cancer, bladder cancer, prostate cancer	3	NCT04534075	Active, recruiting

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
