# Peer review of "The Local Microbiome in Esophageal Cancer and Treatment Response: A Review of Emerging Data and Future Directions"

_cancers, 2023, doi:10.3390/cancers15143562_

Round 1
Reviewer 1 Report
The review article by Pandey et al. discusses the current state of knowledge of the microbiome and esophageal cancer, and highlights research in other cancer types that may be relevant to esophageal cancer. The review is well written and clear, though it focuses more on other cancers than esophageal cancer - this may be understandable given the relative dearth of studies on the local esophageal microbiome and esophageal cancer. However, many of the other studies included in the review (i.e. all the ICI studies) focus on the gut microbiome, and the review may benefit from addressing possible connection of the gut microbiome with esophageal cancer, rather than only the local esophageal microbiome. Aside from this, the review is a helpful addition to the literature. Detailed comments below.
The section on beta-diversity (page 3) is slightly confusing. Beta-diversity simply represents the difference in microbiome composition between two samples (it has nothing inherently to do with time or space, it just depends which samples are being compared and it is only a relative measure). The sentence “A high B-diversity for an individual would represent a MB that has changed more over time compared to an individual with low B-diversity”, can be modified to: “An individual with repeated samples that had high B-diversity between the samples, would represent a MB that is more volatile over time compared to an individual with low B-diversity between repeated samples.” Similarly, the sentence: “IBD patients have demonstrated higher B-diversity compared to their healthy counterparts” is ambiguous. The meaning could be that IBD patients are more different in their MB amongst each other while non-IBD patients are more similar to each other, OR that IBD patients are different from non-IBD patients. I assume the authors meant the latter, but this should be clarified.
Section 5 – do the authors mean a “high” relative abundance of Streptococcus?
There is a strange paragraph separation at the bottom of page 8.
Page 9 description of Gopalakrishnan results, “A cohort of 19 patients with high abundance of a Faecalibacterium species did not reach median progression-free survival, while a cohort of 20 patients with low abundance had a median PFS of 242 days.” – if responders are enriched for Faecalibacterium, why do patients with “low” Faecalibacterium have longer survival?
Page 9 “Fusubacteria” – typo?
Table 1 and description of studies in Table 1, sometimes it is not exactly clear if a study was in vitro, mice, humans, or a combination thereof. Please make sure this is always clear. For example in section 7.3 (Luu et al study), it is not clear if the improved therapeutic outcome was in humans or mice.
Table 2 – specify in the table or the text that these studies are still in progress (no results reported yet). Additionally, allogeneic stem cell transplant is mentioned in the text but not in Table 2.
Second to last sentence is a fragment: “Within cancer populations, the esophageal tumor microenvironment, tumor-infiltrating immune cells, the local esophageal MB, and local signaling interactions between the TME, MB, and immune cells, as well as longitudinal evolution of the peripheral circulating immune cell response to immunotherapy.”
The authors do not discuss that the gut microbiome should also be studied in relation to esophageal cancer and therapeutic response in esophageal cancer cases. While the gut microbiome is in theory not the focus of the review, the authors bring up many other studies of ICI (metastatic melanoma, NSCLC, etc.) where the local microbiome is not studied, but rather the gut microbiome is found to have an effect (because of the known interactions of the gut microbiome with immune cell populations). Similarly, the gut microbiome may be important in esophageal cancer outcomes, in addition to the local esophageal microbiome (and the gut microbiome is easier to study). This should be mentioned as a research gap as well, to encourage researchers to not only examine the local esophageal microbiome but the gut microbiome too. Related to this, have any studies looked at the gut microbiome in relation to esophageal cancer observationally (either prevalence or incidence?)
Author Response
We appreciate Reviewer's 1 careful and thoughtful input to improve our manuscript. Our responses below:
- The section on beta-diversity (page 3) is slightly confusing. Beta-diversity simply represents the difference in microbiome composition between two samples (it has nothing inherently to do with time or space, it just depends which samples are being compared and it is only a relative measure). The sentence “A high B-diversity for an individual would represent a MB that has changed more over time compared to an individual with low B-diversity”, can be modified to: “An individual with repeated samples that had high B-diversity between the samples, would represent a MB that is more volatile over time compared to an individual with low B-diversity between repeated samples.” Similarly, the sentence: “IBD patients have demonstrated higher B-diversity compared to their healthy counterparts” is ambiguous. The meaning could be that IBD patients are more different in their MB amongst each other while non-IBD patients are more similar to each other, OR that IBD patients are different from non-IBD patients. I assume the authors meant the latter, but this should be clarified.
Response: We have revised these sentences for clarification.
3. Section 5 – do the authors mean a “high” relative abundance of Streptococcus?
Response: Yes, this has been corrected.
4. There is a strange paragraph separation at the bottom of page 8.
Response: The paragraph has been reformatted.
5. Page 9 description of Gopalakrishnan results, “A cohort of 19 patients with high abundance of a Faecalibacterium species did not reach median progression-free survival, while a cohort of 20 patients with low abundance had a median PFS of 242 days.” – if responders are enriched for Faecalibacterium, why do patients with “low” Faecalibacterium have longer survival?
Response: We understand the term “did not reach median progression-free (or overall) survival” to describe a scenario in which more than 50% of patients have not shown evidence of disease progression (or died). Therefore, not having reached median PFS or OS is better than having reached particular length of PFS or OS.
6. Page 9 “Fusubacteria” – typo?
Response: Corrected to Fusobacteria.
7. Table 1 and description of studies in Table 1, sometimes it is not exactly clear if a study was in vitro, mice, humans, or a combination thereof. Please make sure this is always clear. For example in section 7.3 (Luu et al study), it is not clear if the improved therapeutic outcome was in humans or mice.
Response: We amended the table to include a column labeled "Subjects" to clarify if the study was in vitro, mice and/or humans.
8. Table 2 – specify in the table or the text that these studies are still in progress (no results reported yet). Additionally, allogeneic stem cell transplant is mentioned in the text but not in Table 2.
Response: We included a column in Table 2 labeled "Status" and added the trial including FMT to allogeneic stem cell transplant.
9. Second to last sentence is a fragment: “Within cancer populations, the esophageal tumor microenvironment, tumor-infiltrating immune cells, the local esophageal MB, and local signaling interactions between the TME, MB, and immune cells, as well as longitudinal evolution of the peripheral circulating immune cell response to immunotherapy.”
Response: Corrected to the following sentence "The esophageal tumor microenvironment, tumor-infiltrating immune cells, the local esophageal MB, and local signaling interactions between the TME, MB, and immune cells, as well as longitudinal evolution of the peripheral circulating immune cell response to immunotherapy should all be investigated in future studies of esophageal cancers."
10. The authors do not discuss that the gut microbiome should also be studied in relation to esophageal cancer and therapeutic response in esophageal cancer cases. While the gut microbiome is in theory not the focus of the review, the authors bring up many other studies of ICI (metastatic melanoma, NSCLC, etc.) where the local microbiome is not studied, but rather the gut microbiome is found to have an effect (because of the known interactions of the gut microbiome with immune cell populations). Similarly, the gut microbiome may be important in esophageal cancer outcomes, in addition to the local esophageal microbiome (and the gut microbiome is easier to study). This should be mentioned as a research gap as well, to encourage researchers to not only examine the local esophageal microbiome but the gut microbiome too. Related to this, have any studies looked at the gut microbiome in relation to esophageal cancer observationally (either prevalence or incidence?)
Response: We included the following on Page 10.
"While the gut microbiome and fecal sampling may not aid in describing local esophageal processes, several studies mentioned in this review (Table 1) implicate intestinal microbiota as modifying ICI treatment response. With the recent addition of ICIs to the treatment of both locally advanced and metastatic upper GI cancers, these study results may also be applicable in the upper GI setting. The same MB-immune cell interactions present in the gut of ICI-treated melanoma or NSCLC patients may be present in upper GI cancer patients as well."

Reviewer 2 Report
the manuscript is oriented to esophageal cancer and the title refered to upper GI generally. It is a general review, the new knowdledge is not clear
the manuscript is oriented to esophageal cancer and the title refered to upper GI generally. It is a general review, the new knowdledge is not clear and no new data included
Author Response
We appreciate Reviewer 2 pointing out the limitations of this review.
The manuscript is oriented to esophageal cancer and the title refered to upper GI generally. It is a general review, the new knowdledge is not clear
Response: We have retitled the manuscript "The Local Microbiome in Esophageal Cancer and Treatment Response: A Review of Emerging Data and Future Directions." We have highlighted our current understanding of the local microbiome's role in esophageal cancer and in future directions, we discuss novel approaches to interrogate the local microbiome. We hope this will provide readers an updated review of the current literature and encourage further investigation.
Reviewer 3 Report
The manuscript entitled "The local microbiome in upper GI carcinogenesis and treatment response" is a good work done by the authors. They did a good literature survey and well analyzed the obtained data. I still suggest some modifications prior to further consideration.
1. Abstract can be improved by incorporating methodology part and number of articles authors assessed. Some quantitative information such as ranges of values can enhance its attractiveness
2. In the introduction, authors must include information on microbiome composition
3. A schematic diagram must be included to show the data collection, sorting etc
4 . Diagram must be included to indicate the Molecular signaling involved
5. Involvement of toll like signaling pathways may be emphasized in the pathogenesis of gastric cancers
6. Section 7.3 and 7.4 may be expanded more with supporting literature.
7. In tables abbreviations must be avoided and if any is included, they must be expanded in legends
Punctuation errors must be checked thoroughly
Author Response
We appreciate Reviewer 3's thoughtful and constructive suggestions to improve our manuscript.
- Abstract can be improved by incorporating methodology part and number of articles authors assessed. Some quantitative information such as ranges of values can enhance its attractiveness
Response: We included "Design" and "Results" to the Abstract.
2. In the introduction, authors must include information on microbiome composition
Response: We believe the body of our manuscript provides much more meaningful information on the microbiome composition. We devote a section of our review at the beginning to define several terms to familiarize the reader on this topic to provide a foundation.
3. A schematic diagram must be included to show the data collection, sorting etc
Response: Our updated manuscript now includes a schematic diagram (Figure 1) to describe data collection and sorting.
4 . Diagram must be included to indicate the Molecular signaling involved
Response: After much consideration and discussion, we did not include a diagram describing molecular signaling as there appear to be many unrelated mechanisms and pathways that cannot be succinctly described in a diagram. We hope that as our knowledge in this field grows to better illustrate clear molecular signaling in this disease in the future
5. Involvement of toll like signaling pathways may be emphasized in the pathogenesis of gastric cancers
Response: We agreed this is an excellent point and have included this to Section 5 "Esophageal Microbiome"
"A potential mechanism of esophageal carcinogenesis involves toll like receptor (TLR) signaling, which allows for epithelial cells to recognize luminal pathogens and its components such as lipopolysaccharides, flagellins, DNA, and RNA. This triggers an innate immune response and inflammation. Alterations in cytokine and chemokine expression can lead to angiogenesis and potential tumorogenesis. High expression of TLR-5 has been associated with progression from metaplasia/dysplasia to esophageal adenocarcinoma [27]."
6. Section 7.3 and 7.4 may be expanded more with supporting literature.
Response: We expanded Sections 7.3 and 7.4.
7. In tables abbreviations must be avoided and if any is included, they must be expanded in legends
Response: We minimized abbreviations and included a legend with each table.
Round 2
Reviewer 1 Report
I do not see Figure 1, Table 1, or Table 2 in the paper, so these need to be put in. Otherwise the authors have addressed all comments adequately.
Author Response
We have updated the manuscript to include all relevant figures and tables.

Reviewer 2 Report
It is a general review, the new knowdledge is not clear and no new data included
Author Response

(The authors gave the same response as above.)

Reviewer 3 Report
Authors have revised the manuscript in accordance with the comments. I recommend the manuscript for consideration in the journal
Author Response

(The authors gave the same response as above.)
